# Urine Cell-Free MicroRNAs in Localized Prostate Cancer Patients

**DOI:** 10.3390/cancers14102388

**Published:** 2022-05-12

**Authors:** Yoko Koh, Matias A. Bustos, Jamie Moon, Rebecca Gross, Romela Irene Ramos, Suyeon Ryu, Jane Choe, Selena Y. Lin, Warren M. Allen, David L. Krasne, Timothy G. Wilson, Dave S. B. Hoon

**Affiliations:** 1Department of Translational Molecular Medicine, Saint John’s Cancer Institute (SJCI), Providence Saint John’s Health Center (SJHC), Santa Monica, CA 90404, USA; yoko.koh@providence.org (Y.K.); matias.bustos@providence.org (M.A.B.); jamie.moon@providence.org (J.M.); rebecca.gentry@providence.org (R.G.); romela.ramos@providence.org (R.I.R.); 2Department of Urology and Urologic Oncology, Saint John’s Cancer Institute (SJCI), Providence Saint John’s Health Center (SJHC), Santa Monica, CA 90404, USA; jane.choe@providence.org (J.C.); timothy.wilson@providence.org (T.G.W.); 3Genome Sequencing Center, Saint John’s Cancer Institute (SJCI), Providence Saint John’s Health Center (SJHC), Santa Monica, CA 90404, USA; suyeon.ryu@providence.org; 4JBS Science Inc., Doylestown, PA 18902, USA; selenayl@gmail.com; 5Division of Surgical Pathology, Providence Saint John’s Health Center (SJHC), Santa Monica, CA 90404, USA; warren.allen@providence.org (W.M.A.); david.krasne@providence.org (D.L.K.)

**Keywords:** microRNA, cell-free microRNA, urine, plasma, prostate cancer, diagnosis

## Abstract

**Simple Summary:**

Urine cell-free microRNAs (cfmiRs) are promising biomarkers for the detection of prostate cancer (PCa) and may replace or complement prostate-specific antigen screening. This pilot study aims to demonstrate the diagnostic utility of urine cfmiRs for early-stage PCa using a robust microRNA (miR) panel based on next-generation sequencing. We assessed urine, plasma, and formalin-fixed paraffin-embedded tumor tissue samples obtained from patients diagnosed with pT2 PCa. Differentially expressed miRs were found in urine, plasma, and tumor samples obtained from PCa patients. Through bioinformatic analysis, several miRs were found as potential cfmiRs with utility for the detection of PCa. Our results showed that specific cfmiRs in urine samples from PCa patients may have potential utility in the detection of early-stage PCa.

**Abstract:**

Prostate cancer (PCa) is the most common cancer in men. Prostate-specific antigen screening is recommended for the detection of PCa. However, its specificity is limited. Thus, there is a need to find more reliable biomarkers that allow non-invasive screening for early-stage PCa. This study aims to explore urine microRNAs (miRs) as diagnostic biomarkers for PCa. We assessed cell-free miR (cfmiR) profiles of urine and plasma samples from pre- and post-operative PCa patients (*n* = 11) and normal healthy donors (16 urine and 24 plasma) using HTG EdgeSeq miRNA Whole Transcriptome Assay based on next-generation sequencing. Furthermore, tumor-related miRs were detected in formalin-fixed paraffin-embedded tumor tissues obtained from patients with localized PCa. Specific cfmiRs signatures were found in urine samples of localized PCa patients using differential expression analysis. Forty-two cfmiRs that were detected were common to urine, plasma, and tumor samples. These urine cfmiRs may have potential utility in diagnosing early-stage PCa and complementing or improving currently available PCa screening assays. Future studies may validate the findings.

## 1. Introduction

Prostate cancer (PCa) is the most diagnosed cancer, and the second leading cause of cancer death in men in the United States [1]. Prostate-specific antigen (PSA) is a serine protease specific to the prostate. PSA screening plays an important role in the early detection of PCa [2]. However, it may be elevated in conditions other than PCa, including benign prostate hyperplasia (BPH) and acute prostatitis [3]. PSA screening has a low specificity while only achieving an adequate sensitivity [4]. In addition, PCa may be detected even if PSA levels are below the cutoff point [5]. The detection rate of PCa is reported to be around 20% among the cohort with PSA under the cutoff point, including high-grade cancers [5,6,7]. For these reasons, there is a need to develop more reliable biomarkers for the early detection of PCa.

Liquid biopsies have rapidly evolved as minimally invasive methods for the management of various cancers [8]. The major source of liquid biopsies has been blood [9,10]; however, urine is a promising source that enables non-invasive and cost-effective repetitive sampling, making it beneficial in longitudinal patient follow-up [11,12]. Urine assessment in PCa is feasible because the prostate gland releases not only prostate epithelium cells, but also nucleic acids, proteins, and exosomes into the urine [13]. Recent technological advancements in improved molecular detection have made it possible to develop novel urine biomarkers for PCa [14]. 

MicroRNAs (miRs) are small non-coding RNAs of about 18–22 nucleotides in length that epigenetically regulate the mRNA expression of targeted genes [15]. MiR profiles of PCa tissues differ from those observed in other cancer types and normal tissues. In contrast with circulating tumor DNAs (ctDNA) which have a limited half-life [8], miRs are stable in urine and blood [16]. In addition, genomic aberrations in PCa tissues are not of high frequency, thus limiting the utility of using ctDNA, particularly in early-stage PCa. Therefore, a panel of urine cell-free miRs (cfmiRs) has potential to serve as a diagnostic, predictive, and/or prognostic panel of biomarkers for early-stage PCa [17]. Several studies have shown the diagnostic potential of urine cfmiRs in PCa patients [18,19,20]. However, miR assessment in these studies was variable in approaches: patient cohorts with different stages, variable sample collection procedures, different miR extraction procedures, miR assays with variable types of polymerase chain reactions, and different data analyses [21,22]. In some studies, urine was sampled after prostate massage to increase the amount of secretion from the prostate gland; however, this method is considered invasive and not reliable [23,24,25]. In addition, patients with BPH or a negative prostate biopsy were assigned to control cohorts in most studies, which complicated the interpretation of results. BPH does not completely exclude the presence of latent PCa, and negative PCa biopsy may be due to diagnostic inaccuracy [26,27]. Thus, more robust, and universal molecular assays, as well as better study designs are needed to overcome these issues. We have previously demonstrated that specific miRs can be detected in blood and urine samples from melanoma patients using a commercially available next-generation sequencing (NGS)-based assay; HTG EdgeSeq miRNA Whole Transcriptome Assay (WTA) (HTG Molecular Diagnostics, Tucson, AZ, USA) [12]. HTG EdgeSeq miRNA WTA is a platform that allows the quantification of 2083 human miRs using a small amount of biofluids with high specificity and reproducibility [28,29]. 

The study was focused on localized PCa to determine if we could detect cfmiR in urine samples. In this study, we analyzed cfmiR profiles in paired urine and plasma samples obtained from PCa patients using HTG EdgeSeq miRNA WTA. We first determined cfmiRs detected in urine samples. To overcome the problem of controls, we compared miR profiles of samples from patients before and after prostatectomy. We demonstrated the detectability of cfmiRs in urine and plasma samples from early-stage PCa patients and the potential utility of urine cfmiRs as diagnostic biomarkers for PCa.

## 2. Materials and Methods

### 2.1. Study Design

This is a prospective pilot study that followed protocol guidelines approved by SJHC/JWCI IRB (SJCI/JWCI-18-0401) and Western IRB (MORD-RTPCR-0995). All participants provided written informed consent. A total of 11 patients with PCa who underwent a robot-assisted radical prostatectomy (RALP) with or without lymph node dissection by T.G.W. between March 2019 and November 2021 at SJHC were enrolled.

Urine and plasma samples from PCa patients (*n* = 11) diagnosed with pT2 were procured at SJHC/SJCI. Paired plasma and urine samples before and after RALP were collected. PCa patients with paired pre- and post-operative (*n* = 8) and PCa patients with pre-operative (*n* = 3) were used in the analyses. The median duration between sample collection and surgery was 0 days (0–34 days) before RALP and 50 days (43–244 days) after RALP. Serum PSA levels of all patients decreased below detection sensitivity at the time of post-operative collection. In addition, plasma (*n* = 24) and urine (*n* = 16) samples from normal healthy donors (NHD) were collected using the same process as with PCa patients. Formalin-fixed paraffin-embedded (FFPE) tissues were obtained from the division of surgical pathology at SJHC from PCa patients diagnosed with pT2 (*n* = 14) and BPH (*n* = 23) patients who underwent RALP and a robot-assisted simple prostatectomy at SJHC, respectively.

### 2.2. Urine Collection and Nucleic Acid Isolation

Urine samples were collected using a standard sterile 100 mL urine collection cup (Medtronic, Minneapolis, MN, USA). Each urine cup contained 0.05 M, pH 8.0 ethylenediaminetetraacetic acid (Bioworld, Little Rock, AR, USA). Urine was aliquoted and stored at −80 °C until assessed. Total RNA was extracted from a 15 mL urine sample using urine nucleic acid isolation kits followed by an automated nucleic acid isolation system (JBS Science Inc., Doylestown, PA, USA). Briefly, urine samples were thawed, and lysis buffer was added. Samples were incubated with beads and washed with ethanol to extract urine nucleic acids. Urine nucleic acids were quantified according to the manufacturer’s protocol. The isolated nucleic acids in elution buffer were then aliquoted and cryopreserved at −80 °C until needed for assays.

### 2.3. Plasma Collection

All blood samples were collected in Streck tubes (Streck, La Vista, NE, USA). All blood samples were centrifuged at 1600× *g* for 10 min at 10 ℃ immediately, aliquoted, barcoded, and cryopreserved as plasma aliquots at −80 °C. 

### 2.4. Preparation of FFPE Tissue Samples

FFPE tissues of PCa and BPH were cut into 5 µm sections using rotary microtome HM 325 (Thermo Fisher Scientific, Waltham, MA, USA) and the regions of PCa and BPH were micro-dissected with sterile scalpels and dissecting needles, as previously reported [12]. The micro-dissected regions were verified by pathologists according to the hematoxylin and eosin-stained slides (Appendix A).

### 2.5. HTG EdgeSeq miRNA WTA for Urine, Plasma, and Tissue Samples

Nucleic acids extracted from urine samples from NHDs and PCa patients were processed with HTG biofluid lysis buffer without incubation, following the HTG user manual. Plasma samples were thawed and incubated with HTG plasma lysis buffer for 3 h at 50 ℃ at 450 rpm on the Mixer HC (USA Scientific, Ocala, FL, USA). Micro-dissected FFPE tissue samples were incubated with HTG bulk lysis buffer, proteinase K, and denaturalization oil for 3 h at 50 ℃ at 450 rpm on the Mixer HC. Then, all lysed samples were processed on the automated HTG EdgeSeq instrument for probe-capture of 2083 validated human miRs for 20 h. After probe-capture was completed, NGS library preparation, bead clean-up, NGS library quality controls, and NGS library normalization and pooling were performed as previously described [12].

The NGS was performed with Illumina NextSeq 550 or MiSeq platforms following HTG EdgeSeq miRNA WTA instructions. Sequences were assessed with a read length of 1 × 50 base pairs. FASTQ files were generated from raw sequencing data using Illumina BaseSpace bcl2fastq software version 2.2.0 and Illumina Local Run Manager Software version 2.0.0. FASTQ files were analyzed with HTG EdgeSeq Parser software version v5.3.0.7184 to generate raw counts for 2083 miRs per sample [12,30,31,32]. All the samples included in this study passed quality control checks. Each HTG miRNA WTA assay includes negative (CTRL_ANT1, CTRL_ANT2, CTRL_ANT3, CTRL_ANT4, and CTRL_ANT5) and positive (CTRL_miR_positive) miR controls, and 13 mRNA housekeeping genes (*ACTB*, *B2M*, *GAPDH*, *YWHAZ*, *PPIA*, *RNU47*, *RNU75*, *RNY3*, *SNORA66*, *RPL19*, *RPS20*, *RPL27*, and *RSP12*). All the controls were included in addition to the 2083 total miR panel. In all runs, Human Brain Total RNA (Ambion, Inc., Austin, TX, USA) was used as a process control for NGS library preparation but was not sequenced.

### 2.6. Bioinformatic and Biostatistical Analysis

Data analyses were processed and visualized using R version 4.1.0 [33] and packages from the Bioconductor project [34]. Raw gene counts were normalized and transformed using the variance stabilizing transform (VST) method from DESeq2 [35] for downstream visualizations and unsupervised clustering analysis. Differential expression analysis was carried out on raw counts within the DESeq2 framework. DESeq2 analyses and statistical comparisons were performed between (1) NHDs vs. pre-operative urine samples; (2) pre- vs. post-operative urine samples; (3) NHDs vs. pre-operative plasma samples; (4) pre- vs. post-operative plasma samples; and (5) BPH vs. PCa tissue samples. *p*-values (<0.05 significant) were calculated using the Wald test adjusted for multiple hypotheses using the Benjamini-Hochberg method [36]. 

Differential expression was calculated using DESeq2 and only differentially expressed (DE) miRs with a false-discovery rate < 0.05 and absolute log_2_ fold change (log_2_FC) > 1 were included. *T*-test analysis and the one-way ANOVA were performed with R version 4.1.0. A two-sided *p*-value < 0.05 was considered statistically significant. All figures were unified using CorelDraw graphics suite 8X (Corel Corporation, Ottawa, ON, Canada) and/or Adobe Illustrator CC (Adobe Inc., Los Angeles, CA, USA).

## 3. Results

### 3.1. Patient Characteristics

Among 11 PCa patients analyzed, only eight patients had paired pre- and post-operative samples and the remaining three patients had only pre-operative urine and plasma samples. The clinicopathological characteristics are summarized in Table 1. All patients were histologically diagnosed with pT2 PCa. No lymph node or distal metastases were detected pathologically or clinically. Normal plasma samples were analyzed from 24 male NHDs and normal urine samples were analyzed from 16 male NHDs with ages ranging from 21–65 years old. PCa and BPH patients with FFPE tissue samples analyzed were independent from patients assessed for urine and plasma samples.

### 3.2. CfmiRs Detected in Urine Samples of PCa Patients

To determine if miRs can be detected in pT2 PCa patients, we assessed urine, plasma, and FFPE tissue samples using HTG EdgeSeq miRNA WTA. All samples were successfully analyzed and passed the quality control checks. First, we assessed miRs profiles in urine samples (pre- and post-operative PCa patients vs. NHDs). The principal component analysis (PCA) revealed that both pre- and post-operative PCa patients have different cfmiR patterns compared to NHDs (Figure 1A). Volcano plots were utilized to visualize DE miRs among different cohorts. A total of 449 miRs were found to be differentially expressed in urine samples of pre-operative PCa patients compared to those of NHDs. Of those, 301 and 148 miRs were up- and down-regulated, respectively (Figure 1B). Eighty-nine cfmiRs were differentially expressed significantly (49 up- and 40 down-regulated, Figure 1C) in urine samples of pre- vs. post-operative PCa patients. We identified 25 cfmiRs that were down-regulated in post-operative samples among DE cfmiRs compared to pre-operative PCa and NHDs (Table 2).

### 3.3. CfmiRs Detected in Plasma Samples of PCa Patients

To assess cfmiR profiles present in plasma, similar analyses for urine samples were performed. PCA demonstrated that the cfmiR pattern detected in plasma samples is similar to that of urine samples (pre-, post-operative PCa patients and NHDs; Figure 2A). The visualization of up- and down-regulated DE cfmiRs was studied by volcano plots. We detected 290 DE cfmiRs between pre-operative PCa patients and NHDs and two cfmiRs between pre- and post-operative PCa patients (Figure 2B,C). Four cfmiRs that were differentially expressed between pre-operative PCa patients and NHD samples were also down-regulated in post-operative PCa patients (Table 3).

### 3.4. Identification of miRs as Diagnostic Markers for PCa

To verify if miR expression profiles detected in plasma and urine samples were present in tumor tissues, we compared the different sources. Tumor-related miRs were identified by comparing tissue miR profiles of PCa and BPH patients. The volcano plot shows DE miRs in PCa tissue samples, of 106 were up-regulated and 189 were down-regulated (Figure 3). Next, we assessed the commonly expressed DE miRs among different sample cohorts (urine, plasma, and tissue) to determine key miRs candidates that may have diagnostic potential. A comparison of tumor-related miRs and DE cfmiRs in urine and plasma between pre-operative PCa patients and NHDs was performed (Figure 4). The results demonstrated that 17.0% (92 of 542) of urine cfmiRs were also found commonly in PCa tissues. Among plasma cfmiRs, 20.2% (108 of 535) were commonly detected in PCa tissue samples. A total of 42 miRs were common in all three comparisons as shown in the Venn diagram (Figure 4) and listed in Table 4.

## 4. Discussion

Liquid biopsy, as a less invasive and practicably repeatable diagnostic tool, is an emerging resource for precision oncology to replace or be combined with conventional tissue-based biopsy [8]. Liquid biopsy provides molecular characterization on circulating tumor cells (CTC), ctDNA, extracellular vesicles, cell-free RNA and miR in different biofluids [9]. MiR profiles have been associated with specific tumor characteristics reflecting each stage of tumor evolution [37]. Compared to ctDNAs and CTCs, miRs have the advantage of abundance and detectability in clinical specimens [38], and may represent better blood biomarkers for diagnosing early-stage cancers. CfmiRs detected in the blood are highly stable and do not rapidly degrade after blood has been drawn as ctDNA does [14]. To date, there are more than 2000 characterized human miRs in the miRbase, v22 [39]. The utility of cfmiRs as biomarkers has gained recent attention due to major technical improvements in the field and the need for non-invasive and cost-effective assays that allow repetitive surveillance and assessment. Although several studies have demonstrated that single miRs or miR signatures have potential utility to serve as diagnostic biomarkers in various types of cancer, there has been no consistent or validated agreement in cfmiRs profiles as diagnostic biomarkers [40]. Thus, there is a need to find novel and robust assays which enable comprehensive profiling of cfmiRs. The HTG EdgeSeq miRNA WTA provides the opportunity to detect 2083 human miRs based on NGS technologies, which is more accurate and sensitive than polymerase chain reaction assays [12]. NGS assays of bulk mRNA/miR sequencing have inherent bioinformatic complexities in defining specific miRs.

Urine represents a valuable source for exploring biomarkers related to PCa because it contains cell-free nucleic acids and cells released from the prostate gland [13,14]. Urine tests have the advantage over blood tests in aspects of non-invasiveness, convenience, and compliance. In this study, HTG EdgeSeq miRNA WTA afforded robust and sensitive detection of cfmiRs in urine, equivalent to those in plasma. Using this novel assay, we demonstrated the detectability of miRs not only in urine but also in plasma and FFPE samples. Several reports have examined the diagnostic value of urine cfmiRs for PCa. However, they have problems with adopting inappropriate control cohorts or having inaccuracy that might detect miRs unrelated to PCa tumors. To overcome these matters, we compared the miR profiles between pre- and post-operative samples. However, DE miRs before and after surgery may include cfmiRs related to the normal prostate gland. To solve the previous problems, we identified the tumor-related miR profiles by comparing miRs between PCa and BPH FFPE tissues and then extracted only cfmiRs detected in PCa tumors. This analysis was possible because HTG EdgeSeq miRNA WTA could identify comprehensive miR profiles accurately with a small amount of fluid samples. The urine assay may have an advantage for early detection, particularly of localized PCa, whereas in systemic disease spreading, blood assessment may be more beneficial. This will have to be investigated in the future.

In this study, we demonstrated that we could detect cfmiRs from urine samples of early-stage localized PCa using an NGS-based assay, HTG EdgeSeq miRNA WTA. Furthermore, by analyzing and comparing commonly DE miRs from different cohorts, we determined key potential miR candidates related to PCa. Our study demonstrated 42 urine cfmiRs that have potential as diagnostic biomarkers for PCa. Among these 42 cfmiRs, several miRs have already been reported to be associated with PCa. These studies are preliminary and functional analyses examining whether miRs have promoting or suppressing effects on tumor required further investigation. The miRs cover a wide range of pathways and mechanisms including those related to tumor progression and early development [41,42,43,44,45]. In previous studies, miR-24-3p, miR-27a-3p, miR-146b-5p, and miR-23a-3p were reported as oncomiRs with the potential to influence tumor cell migration, invasion, and proliferation in PCa cell lines [46,47,48,49,50]. MiR-30a-5p, miR-455-3p, miR-23b-3p, miR-494-3p, and miR-27b-3p were reported as suppressors of specific genes in PCa [42,49,51,52,53]. MiRs are involved in multiple processes of development and progression in cancer, and their expression changes dynamically [17,54]. The remaining miRs have not been explored in PCa to date and may be promising candidates as biomarkers. For further verification of specific miRs, it is necessary to analyze larger cohorts of patients to determine consistency and to include samples from PCa patients at different stages.

This study has several limitations. The main limitation is that the sample size of PCa patients is small and only pT2 PCa patients were included. Thus, further analyses including larger cohorts of different stages of PCa are needed to validate the findings. The other issue that needs to be addressed is the diurnal release of miR into the urine. In this prospective pilot study, we could not establish a uniform timing of sample collections due to patient scheduling. It remains to be determined if the collection time of urine influences the detection levels of cfmiR. This must be further explored in future studies with prospective study designs involving time-course analysis. Fluid diet intake and non-cancer medical conditions could influence urinary frequency. Patients with high and low urination may influence the levels of cfmiRs.

## 5. Conclusions

In this pilot study, we presented a comprehensive cfmiR profiling in urine, plasma, and FFPE tissue samples from PCa patients using HTG EdgeSeq miRNA WTA. We identified 42 urine cfmiRs as potential diagnostic biomarkers for PCa. Further investigations are needed to validate the utility of these cfmiRs for diagnosing PCa.

## Figures and Tables

**Figure 1 cancers-14-02388-f001:**
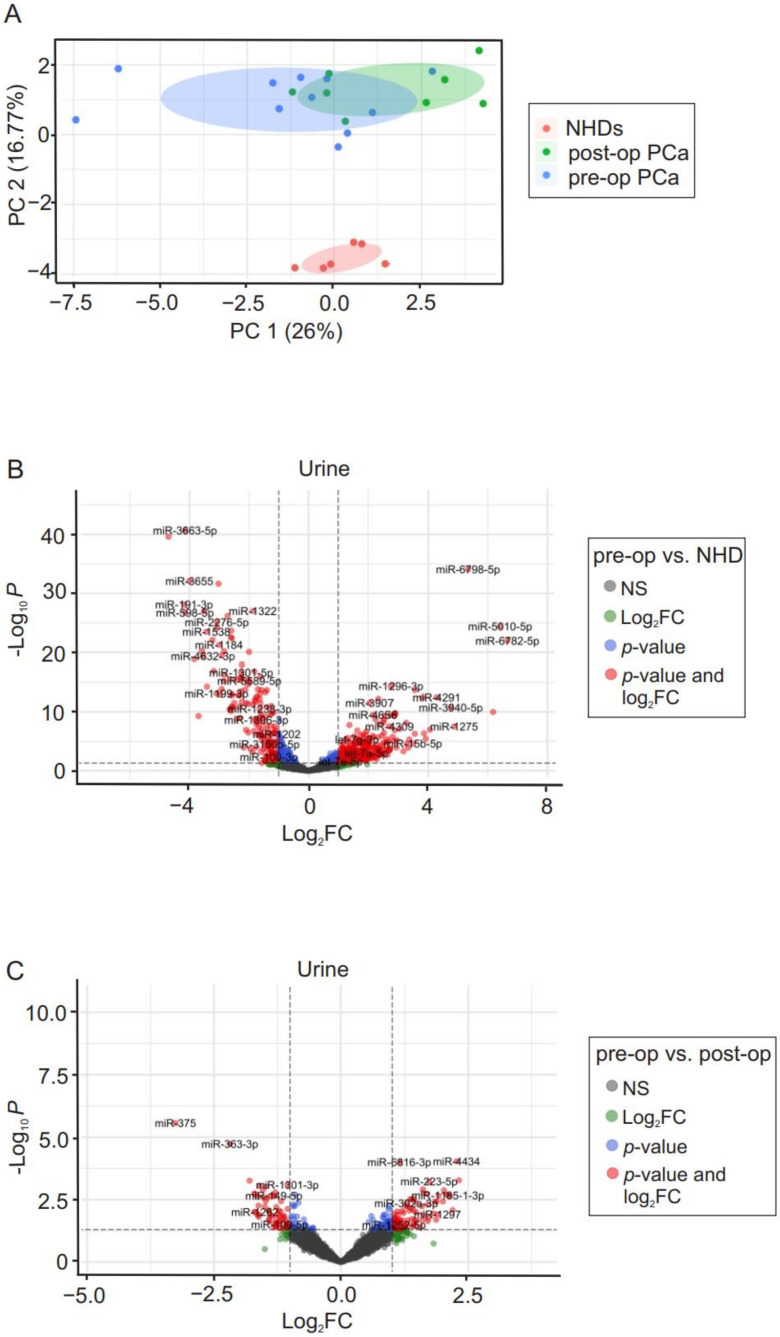
Differentially expressed (DE) cell-free microRNAs (cfmiRs) in urine from PCa patients. (**A**) The principal component analysis (PCA) of microRNAs (miRs) detected in urine samples. The scatter plot of PCA axis 1 (PC1) and axis 2 (PC2) shows the pattern of cfmiR detected in urine samples of normal healthy donors (NHDs), post-, and pre-operative PCa patients. Individual data points are colored accordingly, NHD (pink), post- (green) and pre-operative (blue) PCa patients. (**B**,**C**) Volcano plots of DE miRs in urine samples of pre-operative PCa patients vs. NHDs (**B**) and pre- vs. post-operative PCa patients (**C**) are shown, discriminated based on *p*-value and log_2_ fold change (log_2_FC) at an α level of 0.05 and absolute log_2_FC cutoff of 1. Colored dots correspond to each miR whose expression differences were significant based on both *p*-value and log_2_FC values (red dots), only *p*-value (blue dots), only log_2_FC (green dots), or not significant (grey dots). Down-regulated cfmiRs are on the left side and up-regulated cfmiRs are on the right side (total variables = 2083). NS, not significant.

**Figure 2 cancers-14-02388-f002:**
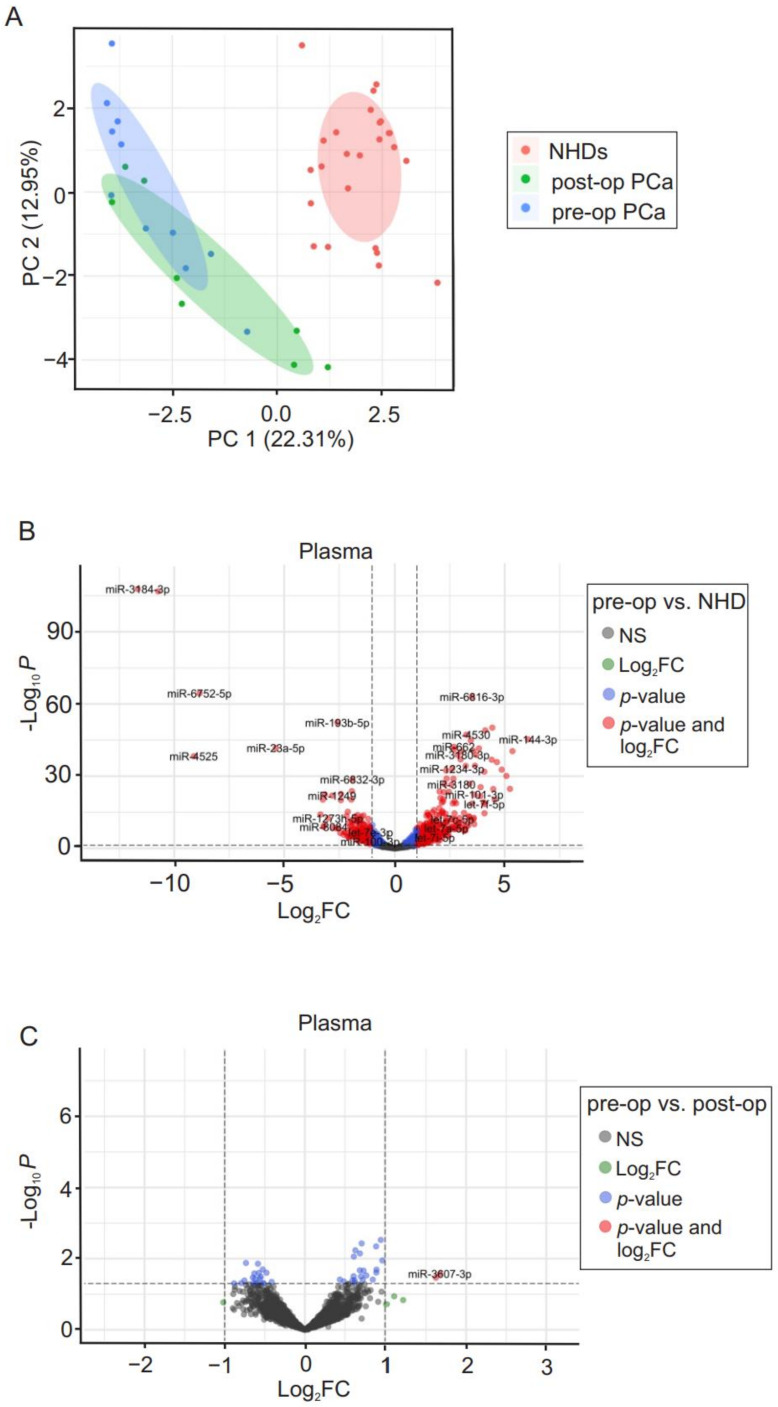
Specific cfmiRs detected in plasma samples from PCa patients. (**A**) The cfmiRs were detected in plasma samples of PCa patients. The scatter plot of PCA axis 1 (PC1) and axis 2 (PC2) shows the pattern of cfmiR detected in plasma samples of NHDs, post-, and pre-operative PCa patients. Individual data points are colored accordingly, NHD (pink), post- (green) and pre-operative (blue) PCa patients. (**B**,**C**) Volcano plots of DE microRNAs in plasma samples of pre-operative PCa patients vs. NHDs (**B**) and pre- vs. post-operative PCa patients (**C**) are shown, discriminated based on *p*-value and log_2_FC at an α level of 0.05 and absolute log_2_FC cutoff of 1. Colored dots correspond to each cfmiR whose expression differences were significant based on both *p*-value and log_2_FC values (red dots), only *p*-value (blue dots), only log_2_FC (green dots), or not significant (grey dots). Down-regulated cfmiRs are on the left side and up-regulated cfmiRs are on the right side (total variables = 2083).

**Figure 3 cancers-14-02388-f003:**
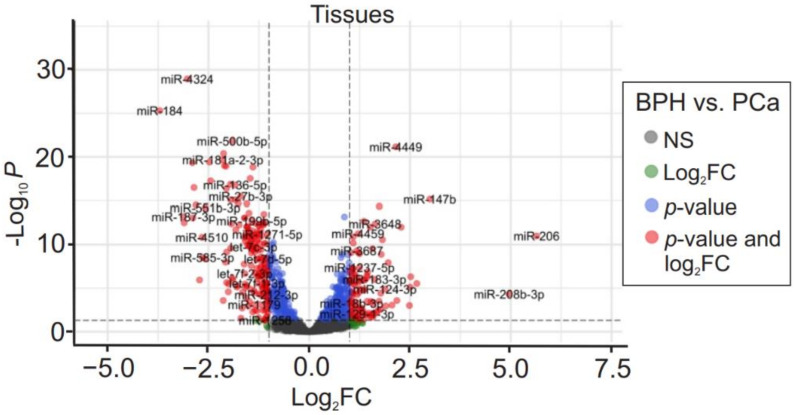
Specific microRNAs (miRs) detected in PCa and BPH tissues. The volcano plot of DE miRs in tissue samples of BPH vs. PCa is based on *p*-value and log_2_FC at an α level of 0.05 and absolute log_2_FC cutoff of 1. Colored dots correspond to each miR whose expression differences were significant based on both *p*-value and log_2_FC values (red dots), only *p*-value (blue dots), only log_2_FC (green dots), or not significant (grey dots). Down-regulated miRs are on the left side and up-regulated genes are on the right side (total variables = 2083).

**Figure 4 cancers-14-02388-f004:**
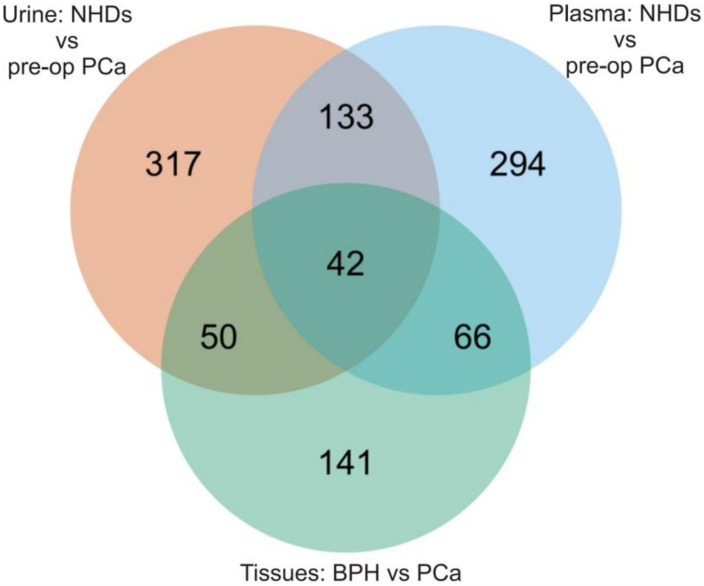
Overlapping DE miRs detected among plasma, urine, and FFPE tissue samples. The Venn diagram shows the significant DE miRs that overlap in all types of samples analyzed.

**Table 1 cancers-14-02388-t001:** Clinicopathological characteristics of prostate cancer (PCa) and benign prostate hyperplasia (BPH) patients.

Variables		PCa with Pre- & Post-op Samples (*n* = 8)	PCa with Pre-op Samples (*n* = 11)	PCa with FFPE Samples (*n* = 14)	BPH with FFPE Samples (*n* = 23)
Age, median (range)		68 (56–76)	69 (56–76)	69.5 (58–75)	68 (51–85)
Serum PSA levels (ng/mL), median (range)		8.75 (4.30–11.20)	8.60 (4.30–11.98)	7.5 (4.5–90.6)	-
Pathological T stage, *n* (%)					
	pT2	8 (100)	11 (100)	14 (100)	-
Grade group, *n* (%)					
	1	1 (12.5)	1 (9.1)	1 (7.1)	-
	2	3 (37.5)	4 (36.4)	0 (0)	-
	3	3 (37.5)	4 (36.4)	6 (42.9)	-
	4	0 (0)	1 (9.1)	5 (35.7)	-
	5	1 (12.5)	1 (9.1)	2 (14.3)	-
Lymphovascular invasion, *n* (%)					
	Present	0 (0)	0 (0)	3 (21.4)	-
	Absent	8 (100)	11 (100)	11 (78.6)	-
Focality, *n* (%)					
	Unifocal	2 (25.0)	4 (36.4)	0 (0)	-
	Multifocal	6 (75.0)	7 (63.6)	14 (100)	-
Surgical margins status, *n* (%)					
	Negative	6 (75.0)	7 (63.6)	11 (78.6)	-
	Positive	2 (25.0)	4 (36.4)	3 (21.4)	-
Biochemical recurrence, *n* (%)					
	Present	0 (0)	0 (0)	3 (21.4)	-
	Absent	8 (100)	11 (100)	11 (78.6)	-

PCa, prostate cancer; BPH, benign prostate hyperplasia; FFPE, formalin-fixed paraffin-embedded; PSA, prostate specific antigen.

**Table 2 cancers-14-02388-t002:** Urine cfmiRs down-regulated in post-operative samples that were also differentially expressed between pre-operative PCa and NHD samples.

miR	log_2_FC	*p*-Value	Adjusted *p*-Value
miR-1262	2.40	0.0006	0.0033
miR-141-3p	1.79	0.0002	0.0013
miR-146a-5p	3.16	0.0005	0.0027
miR-193b-3p	1.38	0.0290	0.0721
miR-200a-3p	2.23	0.0015	0.0066
miR-200a-5p	1.62	0.0024	0.0097
miR-200b-5p	2.18	0.0004	0.0022
miR-200c-3p	2.94	6.65 × 10^−6^	6.22 × 10^−5^
miR-205-3p	1.60	0.0100	0.0308
miR-22-3p	2.01	0.0040	0.0147
miR-29a-3p	3.85	4.56 × 10^−7^	5.36 × 10^−6^
miR-29b-2-5p	1.92	0.0023	0.0094
miR-29b-3p	4.06	1.01 × 10^−7^	1.33 × 10^−6^
miR-29c-3p	3.55	5.36 × 10^−7^	6.17 × 10^−6^
miR-29c-5p	2.13	0.0021	0.0089
miR-30d-5p	1.53	0.0032	0.0124
miR-31-5p	1.33	0.0048	0.0169
miR-363-3p	1.81	0.0039	0.0145
miR-3687	2.93	0.0001	0.0008
miR-4417	2.17	0.0025	0.0101
miR-4709-3p	2.12	0.0059	0.0199
miR-584-5p	1.77	0.0133	0.0379
miR-622	1.76	0.0033	0.0126
miR-663b	2.60	0.0003	0.0015
miR-934	1.92	0.0140	0.0398

cfmiR, cell-free microRNA; PCa, prostate cancer; NHDs, normal healthy donors; miR, microRNA; log_2_FC, log_2_ fold change. *p*-values were calculated using the Wald test adjusted for multiple hypotheses using the Benjamini-Hochberg method.

**Table 3 cancers-14-02388-t003:** Plasma cfmiRs down-regulated in post-operative plasma samples that were also differentially expressed between pre-operative and NHDs plasma samples.

miR	log_2_FC	*p*-Value	Adjusted *p*-Value
miR-1283	1.29	0.0067	0.0354
miR-3692-5p	1.21	0.0072	0.0368
miR-515-5p	1.59	0.0061	0.0329
miR-5187-3p	1.50	0.0070	NA

cfmiR, cell-free microRNA; NHDs, normal healthy donors; miR, microRNA; log_2_FC, log_2_ fold change; NA, not available. *p*-values were calculated using the Wald test adjusted for multiple hypotheses using the Benjamini-Hochberg method.

**Table 4 cancers-14-02388-t004:** The common DE miRs in urine, plasma, and tissue samples of PCa patients.

let-7d-5p	miR-224-5p	miR-362-5p	miR-4751	miR-6802-5p
let-7e-5p	miR-22-5p	miR-3648	miR-487b-3p	miR-6803-5p
miR-1468-5p	miR-23a-3p	miR-377-3p	miR-494-3p	
miR-146b-5p	miR-23b-3p	miR-3937	miR-500a-5p	
miR-147b	miR-24-3p	miR-3943	miR-505-5p	
miR-181a-5p	miR-27a-3p	miR-4271	miR-551b-3p	
miR-181d-5p	miR-27b-3p	miR-4279	miR-561-3p	
miR-186-5p	miR-29b-3p	miR-4417	miR-6738-5p	
miR-221-5p	miR-30a-5p	miR-455-3p	miR-675-3p	
miR-222-3p	miR-335-5p	miR-4638-5p	miR-6789-5p	

DE, differentially expressed; miR, microRNA; PCa, prostate cancer.

## Data Availability

The data will be available upon reasonable requests to Matias A. Bustos.

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
