# Peer review of "Urine Cell-Free MicroRNAs in Localized Prostate Cancer Patients"

_cancers, 2022, doi:10.3390/cancers14102388_

Round 1
Reviewer 1 Report
Thank you for the opportunity to review the manuscript: “MicroRNA without urine cells in patients with localized prostate cancer."
Y.Koh et al presented the results of a pilot study in which they assessed 25 miR (cfmiR) profiles of urine and plasma samples - and postoperative PCa patients and normal healthy donors (using the HTG EdgeSeq miRNA Whole Transcriptome Assay based on NGS.
General comments:
The article is well organized and clear. The subject is of interest to the field.
The methods are correctly described and applied.
The results are clearly presented, and tables and figures are appropriate.
The discussion is balanced and significant.
The conclusions are consistent with the results and discussion.
References are adequate in number and range.
Some comments:
- introduction - lines - 44-45 - The authors should emphasize the presence of prostate cancer with low PSA serum values - this is now a greater diagnostic problem than the increased value of PSA in other diseases (BPH or prostatitis).
- Does a large discrepancy (43-244 days) in the time of taking postoperative samples have an influence on the obtained results?
-
Author Response
Some comments:
- introduction - lines - 44-45 - The authors should emphasize the presence of prostate cancer with low PSA serum values - this is now a greater diagnostic problem than the increased value of PSA in other diseases (BPH or prostatitis).
Response: Thank you very much for your recommendation. This is a very important aspect of the PSA-based diagnostic system. We added more information on the detection rate of prostate cancer with low serum PSA levels and discussed accordingly (p. 2, lines 46-48). Patients screened for prostate cancer having low serum PSA levels are not well diagnosed. There is a need to detect siginificant prostate cancer in early stages. We believe our study provides an approach to improve accurate diagnosis by cfmiR profiling that will be validated in a larger cohort of patients in the future. This miR profile will improve clinical diagnosis by targeting early-stage prostate cancer alone or inconjunction with serum PSA. This would help overcome the problem of under diagnosis of early-stage prostate cancer with low serum PSA levels. At the same time the cfmiR profiles of blood may help in better discerning patients with BPH or prostatitis that have high serum PSA levels.
- Does a large discrepancy (43-244 days) in the time of taking postoperative samples have an influence on the obtained results?
Response: You have asked an important question. In the retrospective setting of this study, sampling of specimens may not be successful due to variable timing of clinic visit or temporary postoperative dysuria. The mechanism of cfmiR turnover is not fully understood. There are studies reporting the half lifetime of cfmiRs is within few days, thus periods of at least 7 weeks are sufficient for postoperative sampling for determining cfmiRs related to true prostate cancer. In future studies, we will perform serial assessment of urine miRs over a defined period post-operation.

Reviewer 2 Report
In the manuscript titled ‘Urine cell-free MicroRNAs in localized Prostate cancer Patients’ authors are trying to establish a standardized non-invasive method using urine cfmiR to detect prostate cancer. They first identified cfmiR signatures in various PCa and non-cancer samples using a NGS-based technique that allowed them to identify multiple cfmiR and then identified 42 DE miR overlapping in all PCa samples.
They did mention that the uniform timing of sample collection could not be established and is currently unknown if there is major variation in the miR secretion based on time, comorbidities, and cancer stage.
Only Concern not addressed is below:
I am unsure why there were different sample sizes ? eg. 11 with 8 (pre and post op), 14 FFPE tumor samples and healthy of 24 plasma vs 16 urine .
1a. Why 14 PCa diagnosed FFPE tissue samples from different people were chosen? Why not use the same patients? I would assume comparing cfmiR in the PCa tissue vs urine of same patient may yield more specific results.
1b. Is the 16 subsets of 24 or separate people?
1b. Was it because of 24 healthy donors they could identify cfmiR in plasma of all but only for 16 in urine?
The study serves as proof of concept and should be scaled further which they intend to do. I congratulate the authors on their work.
Author Response
Only Concern not addressed is below:
I am unsure why there were different sample sizes? eg. 11 with 8 (pre and post op), 14 FFPE tumor samples and healthy of 24 plasma vs 16 urine.
- Why 14 PCa diagnosed FFPE tissue samples from different people were chosen? Why not use the same patients? I would assume comparing cfmiR in the PCa tissue vs urine of same patient may yield more specific results.
Response: We are aware of this issue. However, we already had miR data of prostate cancer FFPE tissue samples. The aim of this study is to establish diagnostic cfmR biomarkers for prostate cancer. The same patients having paired urine, plasma and tissue samples were not available, thus we assessed tissue specimens already analyzed for miR. As this is a pilot study, we believe that the impact of using different cohorts is minimal as it allows a wide spectrum of tissue analysis. In future verification study, paired body fluid and FFPE tissues will be assessed together.
- Is the 16 subsets of 24 or separate people?
- Was it because of 24 healthy donors they could identify cfmiR in plasma of all but only for 16 in urine?
Response: Matched urine and plasma from normal healthy donors (NHDs) were not procured from the same people. The main reason is that plasma samples were initially procured prior to urine collection. analyzed to demonstrate assay feasibility. For this pilot study the primary focus is on the urine assay. Based on the results from blood assay, we have decided to explore the assay on urine samples. Unfortunately, none of the NHDs assessed had paired urine and plasma samples. We will in the future incorporate paired urine and plasma samples strongly consider your suggestion in futureour next study design.

Reviewer 3 Report
The manuscript titled “Urine Cell-Free MicroRNAs in localized Prostate Cancer Patients” is a prospective pilot study in which the authors wanted to find reliable cell-free miRNA as biomarkers from urine for convenient diagnosis of the early stage of prostate cancer without invasion. However, all the important figures are very hard to read with ultra-low resolution. The followings are some concerns and comments have been pointed out that the authors may want to consider updating the manuscript.
Concerns and Comments:
- Line 35 Keywords: I’d suggest the authors switch keywords “urine microRNA” and “plasma microRNA” to others since none of them appear in the main context.
- Line 104 and line 118: I’d suggest the authors provide some details in the methods section instead of most of them were referred to the previous publications. For example, provide relatively detailed nucleic acid/miR isolation procedures since it is one of the major methods of this study.
- Line 119: List the instrument used for cutting the tissues.
- Line 122: Provide the images of hematoxylin and eosin-stained slides.
- Line 158: Please use italic p as it refers to a p-value. Check throughout the whole manuscript.
- Figure 1, Figure 2, and Figure 3: are hard to read, I could not check the details of them. Please provide higher resolution images.
- Line 171 Table 1: Please define R0 and R1 at the bottom of the table. “R0 resection indicates a microscopically margin-negative resection, in which no gross or microscopic tumor remains in the primary tumor bed. R1 resection indicates the removal of all macroscopic disease, but microscopic margins are positive for tumor.”
- Line 184: Please use “89” instead of “eighty-nine”. The format should be homogenous throughout the manuscript.
- Line 201 Table 2, line 225 Table 3: Please add a note at the bottom of the table: briefly describe how the p-value and q-value were calculated? What’s the method?
- Line 216: Please homogenous the format “axis 1” and “axis two”.
- Line 254 Table 4: Please add a note at the bottom of the table to clarify what is the meaning of the bold miRs.
- The miRNA sequencing results should be verified, for example, qPCR, etc. Please provide related data, if not please explain. At least those most important miRNAs.
Author Response
Concerns and Comments:
- Line 35 Keywords: I’d suggest the authors switch keywords “urine microRNA” and “plasma microRNA” to others since none of them appear in the main context.
Response: We agree with your suggestion. We changed the keywords “urine microRNA” and “plasma microRNA” to “microRNA”, “urine” and “plasma” (p. 1, lines 35-36).
- Line 104 and line 118: I’d suggest the authors provide some details in the methods section instead of most of them were referred to the previous publications. For example, provide relatively detailed nucleic acid/miR isolation procedures since it is one of the major methods of this study.
Response: Thank you for your suggestion. We now have described the details of the procedures (p. 3, lines 114-116, 121-139, and p. 4, lines 140-144, 156-157). Also, we explained the detailed procedure of total nucleic acid isolation from urine samples as recommended. The HTG miRNA WTA assay utilizes direct input of plasma and FFPE tissue section samples without isolating miR, which we have now described the process in more detail.
- Line 119: List the instrument used for cutting the tissues.
Response: We added the information on the microtome, rotary microtome HM 325 (Thermo Fisher Scientific, Waltham, MA, USA), according to the suggestion (p. 3, lines 126-127).
- Line 122: Provide the images of hematoxylin and eosin-stained slides.
Response: Thank you for your recommendation. We have added a new figure for H&E-stained slides for two cases of PCa tissues that were included in the study. The H&E-stained slides were examined by two expert surgical pathologists for prostate cancer, Dr. Krasne (Director of Surgical Pathology at SJHC) and Dr. Allen who have over 30 years of experience. Please refer to Supplementary Figure 1 (p. 3, line 130).
- Line 158: Please use italic p as it refers to a p-value. Check throughout the whole manuscript.
Response: We corrected all p-values as italics (p. 4, lines 168, 174, p. 7, lines 216, 218, 219, p. 10, lines 244, 246, 247, p.11, lines 276, all figures and tables).
- Figure 1, Figure 2, and Figure 3: are hard to read, I could not check the details of them. Please provide higher resolution images.
Response: Thank you for your comment. We modified the images to make them easier to read.
- Line 171 Table 1: Please define R0 and R1 at the bottom of the table. “R0 resection indicates a microscopically margin-negative resection, in which no gross or microscopic tumor remains in the primary tumor bed. R1 resection indicates the removal of all macroscopic disease, but microscopic margins are positive for tumor.”
Response: Thank you for the comment. We modified the Table. We changed “R0” and “R1” to “Negative” and “Positive”, respectively (p. 5, Table 1).
- Line 184: Please use “89” instead of “eighty-nine”. The format should be homogenous throughout the manuscript.
Response: Thank you for your suggestion. We spelled it out because 89 came at the beginning of the sentence. We followed the MDPI Style Guide.
- Line 201 Table 2, line 225 Table 3: Please add a note at the bottom of the table: briefly describe how the p-value and q-value were calculated? What’s the method?
Response: We now have added the methods below the Tables. Since analyses were performed using miRs already accounted for the FDR and selected by FDR, we did not see the necessity to present q-valuesom the Tables (p. 7, Table 2 and p. 10, Table 3). We have left the adjusted p-values with the description of method, which are more stringent than q-values.
- Line 216: Please homogenous the format “axis 1” and “axis two”.
Response: We unified the format of the axis (p. 7, line 212 and p. 10, line 240).
- Line 254 Table 4: Please add a note at the bottom of the table to clarify what is the meaning of the bold miRs.
Response: Thank you for your comment. We apologize for this mistake. We modified the Table 4 (p. 12, Table 4). We did not intend to make those miRs bold.
- The miRNA sequencing results should be verified, for example, qPCR, etc. Please provide related data, if not please explain. At least those most important miRNAs.
Response: Thank you for your recommendation. In this study, we utilized the NGS-based EdgeSeq platform. As mentioned in the manuscript, this assay allows detection of a large number of miRs with high accuracy and reproducibility. In the white paper, the comparison / validation of HTG EdgeSeq miRNA WTA and qPCR showed the Pearson’s coefficient value of 0.93 (White Paper - EdgeSeq miRNA Whole Transcriptome Assay; https://www.htgmolecular.com/assets/htg/docs/HTG_miRNA_WTA_Product_Sheet.pdf). Also, we have demonstrated the utility of this system in our previous studies on melanoma and glioblastoma (Bustos MA, et al. Cancers 2020 (10,12) and Bustos MA, et al. Lab Invest 2022 (10)). Hence, we believe on the validity of the assay without the need for further validation.

Round 2
Reviewer 3 Report
The following comments that the authors should consider and please seriously revise the manuscript again before publication.
- Line 182 Table 1: Please consistent the format with or without space before or after the equal sign “=”. Check throughout the manuscript.
- Line 210 Figure 1 legend: NS, not significant should be defined in the legend. Insert “NS” next to /followed “not significant”. Check throughout the manuscript.
Author Response
Response to Reviewer 3 Comments
We wish to express our appreciation to the Reviewer for his or her insightful comments, which have helped us significantly improve the paper.
- Line 182 Table 1: Please consistent the format with or without space before or after the equal sign “=”. Check throughout the manuscript.
Response: We unified the format with space before and after “=” in the manuscript according to the suggestion (p. 1, line 27, p. 3, lines 96, 98, 99, 102, 105, 106, p. 5, Table 1, p. 7, line 221, p. 10, line 249, p. 11, line 278).
- Line 210 Figure 1 legend: NS, not significant should be defined in the legend. Insert “NS” next to /followed “not significant”. Check throughout the manuscript.
Response: We added abbreviations into the bottom of the Figure 1 legend (p. 7, line 221).
